# Pause characteristics of sentence production in Parkinson's disease: Insights from sentence complexity and length

Fatemeh Mollaei [1,2]*, Huw Evans[1], Alexandra Pool[1]

1 School of Psychology and Clinical Language Sciences, University of Reading, Reading, Berkshire, England, United Kingdom, 2 Centre for Integrative Neuroscience and Neurodynamcis (CINN), University of Reading, Reading, Berkshire, United Kingdom

* f.mollaei@reading.ac.uk

## Abstract

### Purpose

Parkinson's disease (PD) affects forward flow of speech including fluency disruptions in 90% of individuals. One of the main parameters affecting flow and fluency of speech is pause behaviour. However, the precise language characteristics of pauses, including sentence complexity and length, and how they contribute to the fluency disruptions of PD are not fully understood. This study examined how sentence complexity and length affect pause behaviour in PD.

### Method

Seventy-one participants, comprising individuals with PD (n = 32) and neurotypical controls (n = 39), read a speech passage aloud. The number and duration of pauses, categorised by location (between, within sentences), sentence complexity (simple, complex), and sentence length (short, long) were analysed. Cognitive ability, assessed using the Montreal Cognitive Assessment (MoCA), and motor speech deficits (i.e., dysarthria) severity, assessed using a speech perceptual ranking, were evaluated and correlated with pause characteristics.

### Results

Individuals with PD produced significantly more pauses across all categories compared to controls. However, only between-sentence and long-sentence pauses were significantly longer in duration. Pause frequency and duration in both groups were higher in more complex and longer sentences. Significant negative correlations were found between MoCA scores and number of pauses. Significant positive correlations were observed between dysarthria severity and duration of pauses.

**Data availability statement:** The datasets generated and/or analyzed during the present study cannot be shared publicly because of ethical restrictions, as they consist of participants' speech recordings. Data generated at University of Reading are available from the University of Reading, School of Psychology and Clinical Language Sciences Ethics Committee (contact via pclsethcis@reading.ac.uk) and data generated at McGill University are available from McGill University, Faculty of Medicine and Health Sciences Institutional Review Board (contact via irbsec.med@mcgill.ca) for researchers who meet the criteria for access to confidential data.

**Funding:** This work was supported by a start-up package from the School of Psychology and Clinical Language Sciences at University of Reading awarded to Dr. Fatemeh Mollaei (FM). There was no additional external funding received for this study. The funders had no role in study design, data collection and analysis, decision to publish, or preparation of the manuscript.

**Competing interests:** The authors have declared that no competing interests exist.

## Conclusion

These findings suggest that increased cognitive-linguistic demands—indexed by sentence complexity and length—may underlie pausing behaviour and contribute to fluency disruptions in individuals with PD. The results extend previous research by highlighting the potential cognitive-linguistic basis of motor speech dysfunction in PD.

---

## Introduction

Neurodegenerative disorders affect millions of people worldwide, with Parkinson's disease (PD) representing the fastest growing and progressively debilitating condition. PD affects approximately 2% of individuals over the age of 65 [1] and is marked by the degeneration of dopaminergic neurons in the substantia nigra [2]. This dopamine loss disrupts basal ganglia function and impairs motor coordination, leading to hallmark symptoms such as bradykinesia, rigidity, postural instability, tremor, and diminished respiratory support [3]. As no cure currently exists, treatment primarily targets symptoms management and the slowing of disease progression [4].

These motor symptoms often extend to the speech mechanism, resulting in hypokinetic dysarthria, a motor speech disorder characterized by reduced loudness, monopitch, monoloudness, breathy and hoarse vocal quality, and imprecise articulation [5]. Such deficits can compromise communication and negatively impact quality of life. Despite clinical guidelines recommending multidisciplinary management, including speech-language therapy (SLT) [6], only a small proportion (3–4%) of individuals with PD receive SLT services, even though up to 90% develop speech impairments [7]. This underutilization likely reflects a range of systemic barriers, including limited resources, late-stage referrals, and a lack of standardized criteria for prioritizing SLT care [8,9]. Identifying objective, speech-based markers of impairment could help guide SLT referrals and treatment decisions.

### Speech impairments in Parkinson's disease

Speech in PD is most commonly impaired by hypokinetic dysarthria, which affects phonation, articulation, and prosody. Vocal deficits often stem from vocal fold bowing and incomplete glottal closure [10], while articulation is marked by reduced movement amplitude and velocity [11]. Prosodic deficits include monotone pitch and loudness, as well as atypical speech rate and pausing patterns [12]. These disruptions, coupled with reduced facial expressiveness and gesture control, contribute to decreased speech intelligibility—particularly in later stages of the disease [13]. Although the precise relationship between disease severity and motor speech impairment remains inconclusive [14], most research suggests a progressive decline in speech intelligibility as PD advances [15].

Speech-language interventions often target respiratory support and speech rate modulation to improve overall fluency [16]. However, these approaches primarily focus on global speech outcomes, and a more detailed understanding of how PD

affects specific components of speech—such as pause behaviour—is still limited. This gap may constrain the development of more targeted and mechanism-based intervention strategies.

## Fluency and pause behaviour in PD

Speech fluency involves the smooth, continuous production of speech, and is shaped by the timing of pauses and modulation of speech rate [17]. In PD, dysfluencies are considered neurogenic, arising from neurological disruption rather than developmental [18].

In PD, speech timing is commonly disrupted, with pause behaviour emerging as a more consistent feature than changes in speech or articulation rate. Speech rate refers to the overall speed of speech production, typically measured as syllables or words per second including pauses, whereas articulation rate reflects the speed of articulatory movements excluding pauses, and pause timing encompasses the frequency, duration, and placement of silent intervals during speech. Speech rate in PD is frequently reported as reduced, although findings are mixed and influenced by task type and disease stage; in some cases, individuals with PD may also exhibit an accelerated or festinating speech rate [19–21]. While articulation rate is often described as relatively preserved, particularly in milder stages, articulatory impairments such as reduced movement amplitude and articulatory undershoot may also be present [22]. Increased frequency and duration of pauses have been observed across a range of speech tasks, including spontaneous speech and reading, and have been associated with reduced speech fluency and executive-linguistic demands [e.g., 19–21, and 23–26]. These findings are further supported by evidence that modulation of basal ganglia–cortical circuits through deep brain stimulation can influence pause characteristics [27, 28:280–283], suggesting that pausing behaviour may reflect disruption of neural systems involved in speech sequencing and timing.

Together, these findings suggest that both higher-level timing processes and segmental articulatory control can be affected in PD, with their relative contributions varying by individual and disease severity. Altered pause duration and distribution are consistently reported and may represent a clinically relevant feature of PD speech. These observations highlight the importance of moving beyond global speech rate measures toward more detailed analyses of pause timing in connected speech, and suggest that interventions targeting planning, prosodic phrasing, and respiratory–speech coordination may be beneficial [29, 30; see also 31].

## Silent pauses and cognitive influence

Although crucial to fluent speech, the role of silent pauses in PD remains underexplored. Early studies offered conflicting results—some found no differences in pause behaviour [32], while others observed more frequent and longer pauses, especially at sentence onsets [33,34]. Silent pauses in PD may reflect more than just motor dysfunction; they may also indicate underlying cognitive-linguistic challenges, such as deficits in lexical retrieval and sentence planning [25].

In healthy individuals, pauses are often placed before complex syntactic structures, where cognitive demand is greatest [35,36]. Older adults also pause more frequently, likely due to declines in working memory and processing efficiency [37]. PD is associated with early cognitive decline, particularly in executive functioning, which includes working memory, planning, and inhibitory control [38,39]. These cognitive deficits likely contribute to dysfluent speech and atypical pausing patterns [40,41].

## Sentence complexity and length in relation to pause behaviour

Few studies have examined how sentence complexity and length influence pause behaviour in PD. In typical speakers, longer and more complex sentences are associated with increased pre-boundary and intra-sentence pauses [42,43]. These pauses may help manage the greater cognitive-linguistic load associated with complex syntax [44]. It remains

unclear whether individuals with PD, who often have both cognitive and respiratory impairments, show increased sensitivity to sentence complexity and length in their pause behaviour.

## Aims and research questions

This study aims to clarify how sentence complexity and length influence speech pausing in individuals with PD. We examined whether PD speakers produce more frequent and longer-duration silent pauses than neurologically healthy controls, and whether these patterns vary based on sentence complexity and length. We also investigated how cognitive function and dysarthria severity interact with linguistic variables in shaping pause behaviour.

Three research questions were addressed:

1. Do speakers with PD produce a greater number and longer duration of silent pauses than control speakers?

2. Does sentence complexity and length affect pause behaviour?

3. How do cognitive ability and dysarthria severity interact with sentence complexity and length to influence pause frequency and duration?

We hypothesized that individuals with PD would produce a higher number of pauses and longer average pause durations compared to controls. We also anticipated that more complex and longer sentences would elicit more frequent and longer pauses in the PD group. Finally, we expected that cognitive ability and dysarthria severity would modulate the effects of sentence complexity and length on pause behaviour. Findings from this study aim to inform SLT assessment and management strategies by identifying linguistic and cognitive markers of fluency disruptions in speech in PD.

## Materials and methods

### Participants

A total of 71 participants were included in the study, divided into two groups: 39 older control participants (OC) and 32 participants diagnosed with idiopathic Parkinson's Disease (PD). The PD group comprised of 23 males (71.88%) and 9 females (28.13%) with a mean age of 67.90 years (SD = 6.47). The mean disease duration from diagnosis was 73.41 months (SD = 62.14), ranging from 3 months to 19 years and 3 months. The OC group consisted of 39 participants, age- and gender-matched to the PD group, with a mean age of 62.71 years (SD = 8.16). Biographical information of participant groups is presented in Table 1. For demographic details please refer to the S1 and S2 Tables.

The data for this study was obtained from pre-collected datasets (PD 1–18) from Mollaei et al. [45] and 46 additional data from two final-year Speech and Language Therapy students at the University of Reading (UoR). Ethical approval

Table 1. Biographical characteristics of participant groups.

| Participant Groups | | |
| --- | --- | --- |
| CHARACTRISTICS | Parkinson's | Controls |
| Age (years) | 67.59 (6.55) | 62.44 (8.23) |
| Gender | 23 M, 9 F | 21 M, 18 F |
| Post 16 Education (years) | 4.72 (2.16) | 3.79 (2.43) |
| MoCA Score | 26.69 (2.40) | 28.36 (1.46) |
| Time Post PD Diagnosis (months) | 73.41 (62.14) | N/A |
| Dysarthria Severity Rating | 55 (13.8) | N/A |

Note: Values shown are means (Standard Deviations), unless stated otherwise; F = Female; M = Male. MoCA = Montreal Cognitive Assessment

was granted by the UoR Research Ethics Committee and the McGill Faculty of Medicine Institutional Review Board. Prior to participation, all individuals provided fully informed written consent, including consent for the anonymized use of their data in subsequent studies. The recruitment period for the UoR portion ran from 10/06/2021–07/03/2023, while data from McGill University were accessed between 10/06/2021 and 10/04/2022.

All participants were instructed to refrain from taking PD medication for at least 12 hours before data collection. Additionally, no participants had other medical or neurological conditions, including uncorrected hearing or visual impairments.

## Study design

A cross-sectional mixed-method design was used. For between-group comparisons, the independent variable was disease status, with two levels: PD and OC. For within-group analyses, the independent variable was sentence type, which included five levels: within-sentence silent pauses (SP) for short sentences, long sentences, simple sentences, complex sentences, and between-sentence silent pauses. Sentence length was operationalised using a relative short vs. long distinction based on the distribution of the stimulus set (Rainbow passage), rather than on an absolute sentence length criterion. The distribution of sentences in the Rainbow passage:

• Short sentences (S): 11 sentences, ranging from 7–17 words (mean: 11.82 words)

• Long sentences (L): 8 sentences, ranging from 21–28 words (mean: 23.75 words)

• Gap between categories: 4 words (the shortest long sentence has 21 words, while the longest short sentence has 17 words)

The 4-word gap demonstrates that there is no overlap between the two categories, which is essential for valid statistical comparison. Fig 1 shows both the distribution of sentences across the passage and a clear range comparison demonstrating this separation.

The categorization of sentence types and passage sentences is presented in Appendices 1 and 2 based on DeDe and Salis [46]. The dependent variables were as follows:

1. Number of pauses per sentence type

2. Duration of within-sentence SP (seconds)

3. Duration of between-sentence SP (seconds)

## Procedure

Each participant completed the Montreal Cognitive Assessment (MoCA) as a short cognitive screen [47]. The MoCA was delivered in person by Mollaei et al. [45], and virtually during the UoR data collection due to COVID-19 adaptations. The standard "Rainbow Passage" [48], a controlled speech sample of 327 words, was provided to participants. The Rainbow Passage consists of 19 sentences, with a mean length of utterance of 17.21 words (range: 8–35 words). In Mollaei et al.'s study, the passage was presented in person, while during the UoR collection, the passage was sent electronically in advance. Participants had the option to print it out or read it on a computer screen.

In Mollaei et al. [45], the passage was read aloud by participants and recorded for later analysis. During the UoR data collection, recordings were captured via Zoom (Zoom Video Communications Inc., 2016). Participants were instructed to sit in a quiet room with minimal distractions, and headphones/earphones were used to enhance sound quality. Task instructions were sent in advance, and participants were asked not to read the passage prior to the session to avoid adaptation effects [49].

At the beginning of each session, the study's purpose and aims were explained, and participants had the opportunity to ask questions. Demographic information was collected using screening questions. The MoCA was administered during

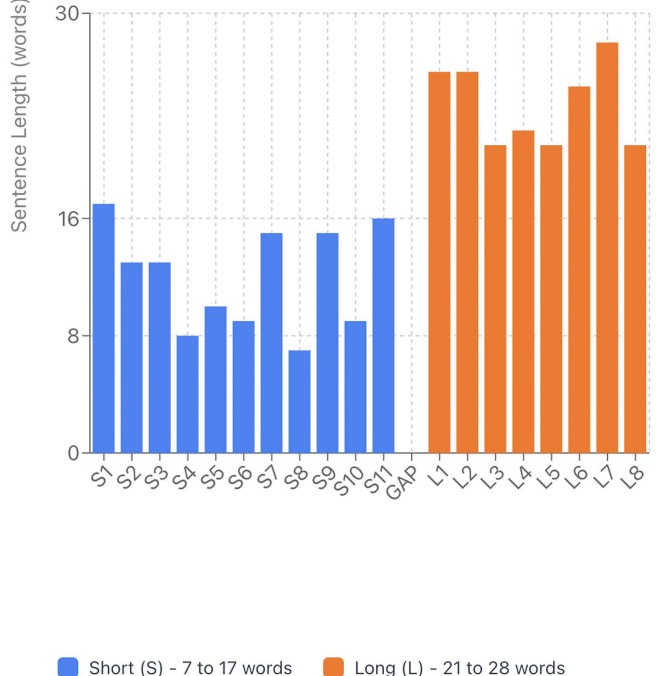

**Short (S) - 7 to 17 words** **Long (L) - 21 to 28 words**

**Fig 1. Distribution of sentence length across the Rainbow Passage.**

the session. Participants were then asked to read the "Rainbow Passage" aloud as they normally would. Audio recordings were taken for analysis.

Audio recordings were saved in WAV format and imported into Praat software (version 6.1.04) [50]. The audio files were visually displayed in Praat, allowing coders to view and listen to the recordings simultaneously. A TextGrid was created in Praat to mark silent pauses, defined as intervals exceeding 150 ms based on Alvar et al. [51]. For this study, pauses were set at a boundary of 200 ms, based on recommendations by DeDe and Salis [46], to capture shorter pauses while excluding those that likely corresponded to breath or articulation. A 200-ms pause duration threshold is commonly justified in studies PD speech because it provides a principled boundary between brief articulatory gaps and functionally meaningful pauses related to speech planning, timing, and cognitive–linguistic processing. Silent intervals shorter than approximately 200 ms are generally considered part of normal articulatory coordination and segmental timing, whereas pauses exceeding this duration are more likely to be perceived by listeners as disruptions to speech flow and to reflect higher-level processes such as lexical retrieval, syntactic planning, or respiratory control [52,53]. In PD, basal ganglia dysfunction is known to affect speech timing and sequencing, making pauses above this threshold particularly relevant as markers of impaired motor planning and cognitive–linguistic integration [27,28]. Using a 200-ms cutoff therefore helps isolate pauses that are sensitive to PD-related speech impairments while excluding micro-silences that are unlikely to be clinically or theoretically meaningful, and this threshold has been widely adopted in analyses of connected speech in PD and related populations [25,52].

The coding system included categories such as silent and filled pauses within-sentence, between-sentence silent and filled pauses, false starts, mazes, revisions, and repetitions (S3 Table provides definitions) based on previous studies [46]. Breaths were treated as silent pauses due to the inability to distinguish individual breath instances. Each sentence was coded individually in the TextGrid to ensure accurate pause placement.

The coded data was exported to Microsoft Excel, and the following measures were extracted for analysis:

 

• Mean duration of between-sentence pauses

• Total number of pauses

• Mean number of pauses per sentence type (short, long, simple, complex)

• Mean duration of pauses per sentence type (short, long, simple, complex)

Passage recordings were additionally analyzed for clinical dysarthria severity, rated by a trained Speech and Language Therapist (SLT) using a 7-point scale (1 = normal speech, 7 = severe speech) based on perceptual characteristics related to articulation, resonance, prosody, phonation, and respiration [54].

While Praat was used to initially identify pauses, each recording was visually inspected by two independent raters, and silence boundaries were manually adjusted as needed to ensure accurate determination of pause onsets and offsets. Inter-rater reliability for these adjustments was assessed using qualitative ratings for both the number and duration of pauses, yielding excellent agreement (reliability coefficients: 0.93 for pause number and 0.86 for pause duration) [55], indicating that measurements were consistent across raters despite the minor manual corrections.

## Statistical analysis

Statistical analyses were conducted using SPSS software, version 28 (IBM Corp., 2021). Descriptive statistics were generated to examine the number and duration of pauses in all sentence types, separately for the PD and OC groups. Descriptive statistics for between-sentence pauses were also computed.

Due to the presence of extreme outliers in some sentence types, these were removed from the analysis: OC25, PD20, and PD27. After removing the outliers, descriptive statistics were recalculated. Although the PD group did not meet normality assumptions (S4 Table), parametric repeated measures ANOVAs were used because they are robust to violations of normality, particularly with moderate sample sizes and balanced repeated measures [56]. Disease status (PD vs. OC) was the independent variable, and the dependent variables were the duration of pauses and the number of pauses, compared across sentence complexity (simple vs. complex) and sentence length (short vs. long).

To explore the effects further, independent-sample t-tests were performed and corrected for multiple comparisons using the Bonferroni correction. For the 10 tests, the Bonferroni-corrected alpha-level was set at 0.005. Paired-sample t-tests were conducted to examine within-subject effects, with the corrected alpha-level for the PD and OC groups set at 0.0125.

Finally, within the PD group, Spearman's correlations were conducted to examine the relationships between (1) dysarthria severity and the number and duration of pauses, and (2) cognitive ability (as measured by the MoCA) and the number and duration of pauses at different sentence levels.

## Results

### Pause number

**Sentence length.**  There was a significant main effect of disease status, with the PD group producing more pauses (mean = 2.185) overall than the OC group (mean = 1.595; see Table 2). A significant effect of sentence length was also observed, with long sentences eliciting more pauses (mean = 2.676) than short sentences (mean = 1.033). The interaction between disease status and sentence length was not significant.

Post-hoc analyses (Bonferroni-corrected) showed that both groups produced significantly more pauses in long than short sentences. Thus, sentence length increased pause number similarly in PD and OC participants (Fig 2).

### Sentence complexity

For sentence complexity, there was again a significant main effect of disease status, with PD participants producing more pauses (mean = 1.938) than controls (mean = 1.404; Table 2). Complex sentences elicited more pauses (mean = 2.224)

**Table 2. Summary of ANOVA and Post-hoc Analysis Results for Pause Characteristics.**

| Analysis | Measure | Sentence Length | | | | Sentence Complexity | | | |
|---|---|---|---|---|---|---|---|---|---|
| | | F/t | df | p | Result | F/t | df | p | Result |
| ANOVA | Disease Status | 11.905 | 1, 66 | <.001*** | PD>OC | 12.123 | 1, 66 | <.001*** | PD>OC |
| Pause Number | Sentence Type | 327.827 | 1, 66 | <.001*** | Long>Short | 231.502 | 1, 66 | <.001*** | Complex>Simple |
| | Interaction | 3.938 | 1, 66 | .051 | n.s. | 6.604 | 1, 66 | .012* | Sig. |
| ANOVA | Disease Status | 6.935 | 1, 66 | .011* | PD>OC | 9.456 | 1, 66 | .003** | PD>OC |
| Pause Duration | Sentence Type | 67.873 | 1, 66 | <.001*** | Long>Short | 15.637 | 1, 66 | <.001*** | Complex>Simple |
| | Interaction | 4.263 | 1, 66 | .043* | Sig. | 0.890 | 1, 66 | .349 | n.s. |
| PD Within-Group | Pause Number | 13.060 | 29 | <.001*** | Long>Short | 5.394 | 29 | <.001*** | Complex>Simple |
| | Pause Duration | 9.548 | 29 | <.001*** | Long>Short | 2.940 | 29 | .006** | Complex>Simple |
| OC Within-Group | Pause Number | 14.742 | 37 | <.001*** | Long>Short | 6.542 | 37 | <.001*** | Complex>Simple |
| | Pause Duration | 12.641 | 37 | <.001*** | Long>Short | 2.517 | 37 | .016 | n.s. |

*Note.* PD = Parkinson's Disease; OC = Older Control; n.s. = not significant; Sig. = significant. *p < .05; **p < .01; ***p < .001. Bonferroni-corrected α = .0125 for within-group comparisons.

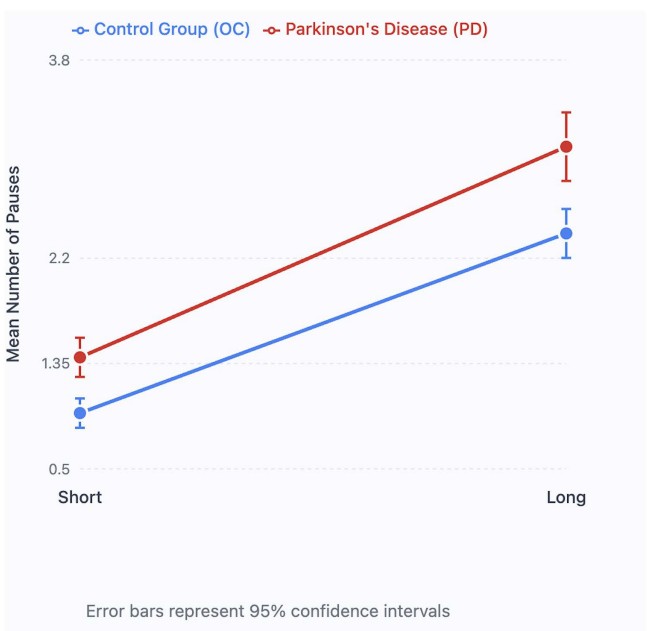

**Fig 2. Pause numbers and sentence length.** Mean number of pauses in short and long sentences in Parkinson's Disease (PD) and Older Healthy (OC) participants.

than simple sentences (mean = 1.118). A significant interaction between disease status and sentence complexity was observed.

Post-hoc analyses indicated that both groups produced more pauses in complex than simple sentences, although this effect was more pronounced in the PD group (Fig 3).

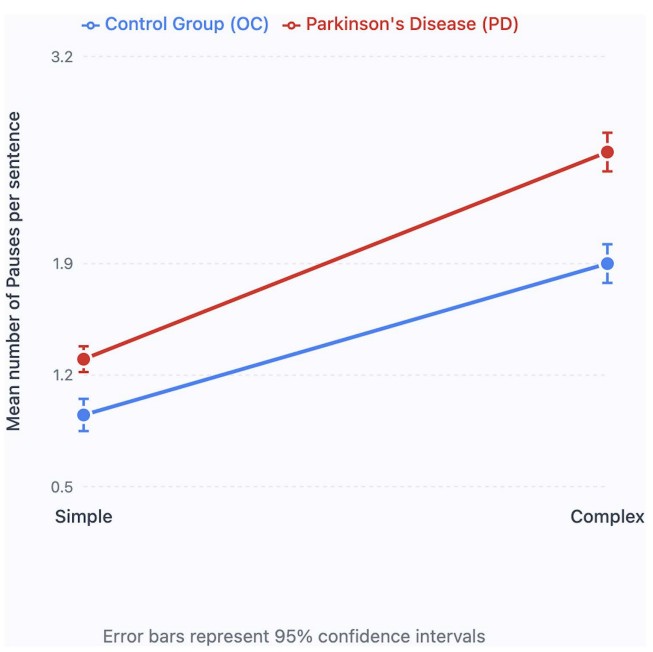

**Fig 3. Pause numbers and sentence complexity.** Mean number of pauses in simple and complex sentences in Parkinson's Disease (PD) and Older Healthy (OC) participants.

## Pause duration

**Sentence length.** A significant main effect of disease status was found for pause duration, with longer pauses in the PD group (mean = 0.396 seconds) compared to controls (mean = 0.345 seconds; Table 2). Sentence length also significantly affected pause duration, with longer sentences associated with longer pauses (mean = 0.411 seconds) compared to short sentences (mean = 0.331 seconds). Importantly, a significant interaction between disease status and sentence length indicated that the effect of length on pause duration differed between groups.

Post-hoc comparisons confirmed that both groups produced longer pauses in long compared to short sentences, with a greater increase observed in the PD group (Fig 4).

## Sentence complexity

For sentence complexity, pause duration was significantly longer in the PD group (mean = 0.408 seconds) than in controls (mean = 0.353 seconds; Table 2), and complex sentences were associated with longer pauses (mean = 0.395 seconds) than simple sentences (mean = 0.365 seconds). However, the interaction between disease status and sentence complexity was not significant.

Within-group analyses showed that the PD group produced significantly longer pauses in complex than simple sentences, whereas this difference did not survive correction in the control group (Fig 5).

## Relationship with cognitive ability

Significant negative correlations in the PD group were observed between cognitive ability, as measured by the MoCA, and the mean number of pauses in both short ($r_s$ (28) = −0.437, $p$ = 0.016) and long sentences ($r_s$ (28) = −0.477, $p$ = 0.008). Greater numbers of pauses were associated with lower cognitive ability. Similar negative correlations were found for the

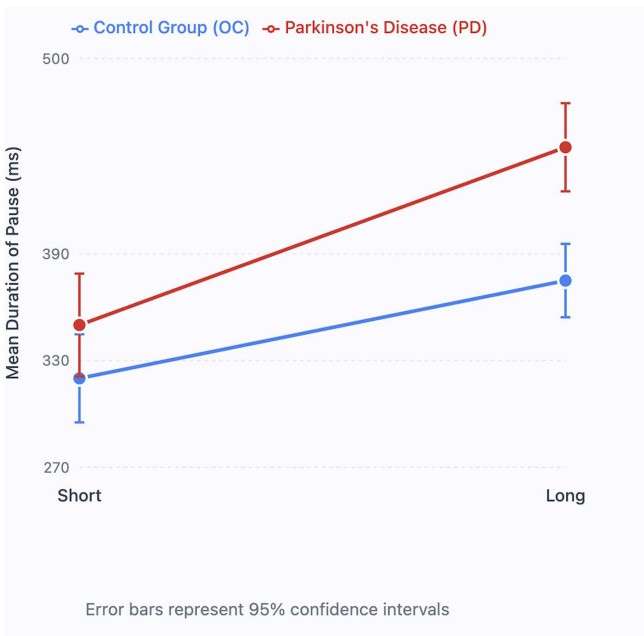

**Fig 4. Pause duration and sentence length.** Mean duration of pauses in short and long sentences in Parkinson's Disease (PD) and Older Healthy (OC) participants.

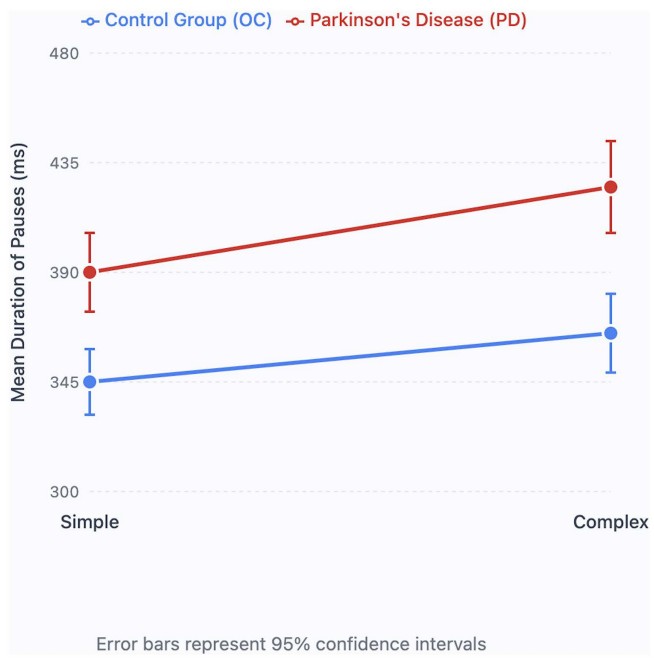

**Fig 5. Pause duration and sentence complexity.** Mean duration of pauses in milliseconds in simple and complex sentences in Parkinson's Disease (PD) and Older Healthy (OC) participants.

mean number of pauses in simple ($r_s$ (28) = −0.436, $p=0.016$) and complex sentences ($r_s$ (28) = −0.492, $p=0.006$), as well as the total number of pauses in the passage ($r_s$ (28) = −0.502, $p=0.005$).

No significant correlations were found between cognitive ability and the duration of pauses in short sentences ($r_s$ (28) = −0.082, $p=0.667$), long sentences ($r_s$ (28) = −0.196, $p=0.300$), simple sentences ($r_s$ (28) = −0.051, $p=0.790$), complex sentences ($r_s$ (28) = −0.187, $p=0.321$), or between-sentence pauses ($r_s$ (28) = −0.273, $p=0.144$).

### Relationship with dysarthria severity

Significant positive correlations were found between dysarthria severity, as measured by the dysarthria rating, and the duration of pauses in short ($r_s$ (28) = 0.420, $p=0.021$), long ($r_s$ (28) = 0.428, $p=0.018$), and complex sentences ($r_s$ (28) = 0.509, $p=0.004$). Longer pauses were associated with greater dysarthria severity. While not statistically significant, a borderline positive correlation was observed between dysarthria severity and the duration of pauses in simple sentences ($r_s$ (28) = 0.359, $p=0.052$).

No significant correlations were found between dysarthria severity and the mean number of pauses in short (Spearman's rho=0.227, p=0.227), long (Spearman's rho=0.035, p=0.854), simple (Spearman's rho=0.055, p=0.772), or complex sentences (Spearman's rho=0.188, p=0.320), nor with the total number of pauses (Spearman's rho=0.114, p=0.548).

## Discussion

### Overview

This study utilized a mixed design to investigate the influence of sentence complexity and sentence length on both the frequency and duration of silent pauses in speakers with Parkinson's Disease (PD) compared to control speakers. It was hypothesized that PD speakers would exhibit an increased number and longer duration of pauses at all sentence levels—short, long, simple, and complex—relative to control speakers. The effect of sentence complexity was expected to influence both the frequency and duration of pauses, with more complex sentences leading to increased pause frequency and duration when compared to simple sentences. Similarly, sentence length was predicted to affect both the frequency and duration of pauses, with longer sentences expected to result in more frequent and prolonged pauses. Cognitive ability and dysarthria severity were anticipated to correlate with both the complexity and length of sentences; however, no specific predictions were made regarding these variables. The main findings from the study are as follows:

1. PD speakers exhibited a greater total number of pauses throughout the speech sample compared to control speakers.

2. PD speakers produced longer durations of between-sentence pauses relative to control speakers.

3. Significant differences between PD and control speakers were found at all sentence levels, except for the mean duration of pauses in short, simple, and complex sentences, as well as the mean number of pauses in simple sentences.

4. Within-group comparisons of sentence complexity revealed that both PD and control speakers produced more pauses in more complex sentences, though this effect was not observed in the duration of pauses across sentence complexities.

5. Sentence length had a significant effect within both groups, with longer sentences producing more pauses and longer durations of pauses in both PD and control groups.

6. Significant negative correlations were found between cognitive ability and the number of pauses in short, long, simple, and complex sentences, as well as the total number of pauses.

7. Significant positive correlations were observed between dysarthria severity and the duration of pauses in short, long, and complex sentences.

The results from this study demonstrate notable differences in the number and duration of silent pauses between PD and control speakers across all sentence levels. As hypothesized, PD speakers exhibited greater durations of between-sentence pauses, and at each sentence level, the frequency and duration of pauses were increased, except for the duration of pauses in short, simple, and complex sentences, as well as the mean number of pauses in simple sentences. The increased total number of pauses in PD speech aligns with prior research indicating that PD speakers require more preparation time to produce speech [23] and tend to produce atypical pauses within sentences [25]. This observation further supports the notion that PD speech is marked by reduced fluency, which may result from a combination of linguistic, motor, and cognitive factors, though the precise interaction between these factors remains unclear. It has been found that pause duration correlates with executive functioning, grammatical ability, and motor initiation, indicating that linguistic planning and executive control — not just motor capacity — influence how long speakers pause [23]. Krivokapić [43] also suggests that sentence length may influence both pre- and post-boundary pause durations, which increases processing effort and planning time. This could explain the observed increase in between-sentence pauses. However, future studies need to investigate linguistic, motor, and cognitive factors to better understand their relative contributions to pause duration.

Another potential explanation for the increased pause duration in PD speakers is the phenomenon of speech initiation hesitation. While this is unlikely to affect group differences, it may explain individual variability in pause duration. Moretti et al. [57] suggested that speech initiation delay, exacerbated by motor symptoms such as gait freezing in PD, may contribute to longer durations of between-sentence pauses. This should be considered when interpreting individual data. Nonetheless, in the present study, we did not examine individual differences in speech initiation difficulties in relation to pause behaviour in PD. Future studies could explicitly address this issue to disentangle initiation-related deficits from pause characteristics and to better understand how initiation difficulties may contribute to pausing in PD.

Interestingly, no significant differences were found in the duration of pauses in short, simple, and complex sentences, nor in the mean number of pauses in simple sentences in PD compared to the OC group. Although many studies report increased pause frequency or duration in PD similar to our findings, some studies have shown reduced or fewer pauses, especially in contexts where speech rate accelerates or rhythmic timing is disrupted [58–60]. For example, PD speakers exhibited fewer overall pauses and longer silent pauses, with a reduction in filled pauses compared to controls, suggesting impaired motor planning and automatic responses in pausing behaviour in PD (*e.g.*, [61]). Similarly, accelerated speech in PD has been associated with reduced total pause time and fewer within-word pauses, indicating that dysregulated pacing and timing can lead to compressed pause patterns (*e.g.*, [59]). These findings are inconsistent with ours. The reason for this discrepancy remains unclear. However, one possible explanation is that uncontrolled acceleration of speech (speech festination) and difficulty terminating ongoing motor actions—both of which may occasionally occur in individuals with PD speech—could lead to compressed speech characterized by fewer, or even absent, planned pauses, particularly at linguistically appropriate boundaries [22,59]. However, we did not observe this pattern in our own data. In such cases, similar or reduced pausing in PD compared to controls may reflect impaired motor pacing rather than increased fluency. Conversely, reduced movement range, impaired respiratory–speech coordination, and increased planning demands may result in more frequent or longer compensatory pauses, especially in longer or more complex utterances [54,62]. The effects of sentence complexity and sentence length on the number and duration of pauses were consistent with predictions, with both factors contributing significantly to the differences observed between PD and control speakers. As expected, increased sentence complexity was associated with more pauses, likely due to the greater cognitive load required to process more complex structures. For PD speakers, this cognitive demand likely results in the production of smaller linguistic units, leading to more frequent pause, compounded by impaired physiological control of respiration and motor control [36]. Together, these findings suggest that pausing in PD reflects a dysregulation of speech timing and motor

control, with observed patterns depending on task demands, sentence characteristics, and individual differences in motor impairment.

The effect of sentence length was as predicted contributing to an increase in both the number and duration of pauses. Constituent length is the linear size of a constituent, measured as the number of words it contains, whereas information load refers to the amount of information or processing demand associated with a constituent. Although these notions are often correlated, they are conceptually distinct and not equivalent. Previous studies have shown that longer sentences are associated with more frequent and prolonged atypical pauses [40,42]. However, our current study did not control for constituent length, measured as syllable count, making it unclear whether the observed effects reflect information load or length-based processing factors. Although these variables are often correlated, they are conceptually distinct, and future work will be necessary to disentangle their individual contributions.

Significant negative correlations were found between cognitive ability and the number of pauses in several sentence categories (short, long, simple, complex, and total number of pauses). Reduced cognitive resources in PD speakers likely lead to smaller linguistic units and an increase in pause frequency. Previous research has shown that individuals with PD exhibit longer and more frequent silent pauses, reduced use of filled pauses, and poorer alignment of respiratory pauses with grammatical boundaries, reflecting disruptions in linguistic planning, automatic cueing, and respiratory–speech coordination [30,63,64]. Much of this evidence comes from spontaneous or semi-spontaneous speech tasks, such as narratives or picture descriptions, which place high demands on lexical retrieval, discourse planning, and internal timing, thereby amplifying pause abnormalities in PD [55,63]. Silent pauses in PD have been associated with increased cognitive–linguistic load and lexical access difficulty, whereas filled pauses—typically serving as planning buffers—are reduced, likely due to basal ganglia dysfunction affecting internally generated cues [23,63]. This study extends this literature by examining pausing during a controlled reading task (Rainbow Passage), in which lexical content and syntactic structure are externally provided, reducing discourse-level planning demands while preserving sentence-level formulation and respiratory coordination requirements. Despite these reduced demands, sentence complexity (simple vs. complex) and sentence length (short vs. long) systematically modulated pause behavior in PD, with longer and more complex sentences eliciting longer silent pauses, particularly within sentences. Although respiratory pauses were not directly measured, the altered distribution of silent pauses within grammatical structures is consistent with prior findings of impaired coordination between respiration and linguistic planning in PD [30]. Together, these findings indicate that pause abnormalities in PD are evident across speech task types and reflect an interaction between motor–respiratory constraints and higher-level linguistic planning, with sentence complexity and length exacerbating these effects even during passage reading.

Dysarthria severity was positively correlated with the duration of pauses in short, long, and complex sentences, with borderline significance in simple sentences. This finding is consistent with previous work by Hammen and Yorkston [61], who reported longer pause durations in speakers with more severe dysarthria and suggested delayed speech initiation as one possible contributing factor. However, increased pause duration may also reflect other mechanisms, including reduced speech rate, increased articulatory or linguistic planning demands, respiratory–phonatory constraints, or more general motor control limitations. Further investigation is therefore warranted, particularly with respect to the relationship between dysarthria severity, pause behavior, and speech rate, which was not controlled for in the present study.

In the present study, we used a relatively short pause duration threshold to capture a broader range of speech interruptions, including brief hesitations that may reflect subtle motor or planning disruptions in PD. Prior research on pausing in speech has varied widely in how pauses are defined. For example, studies of spontaneous speech have commonly used thresholds around 100–250 ms to distinguish short articulatory or motor pauses from longer cognitive–linguistic pauses, with 250 ms often taken as a benchmark for long pauses reflecting planning processes [25,46,65], and some work in PD has used even shorter thresholds (e.g., ~ 15 ms) to enable fine-grained analysis of hesitation patterns [63,66]. Including very short pauses likely increased the number of detected pauses and lowered mean pause durations compared to studies using longer cut-offs. However, because the same threshold was applied consistently across all sentences and

conditions, the observed effects of sentence complexity and sentence length on pausing remain robust. Moreover, using a lower threshold may help detect subtle changes in timing and fluency that are salient in PD but could be overlooked with higher thresholds. Future work should empirically examine how pause threshold selection affects pausing metrics in PD and considers reporting multiple pause categories to facilitate comparability across studies.

## Limitations and future directions

Several limitations should be considered when interpreting the results of this study. The lack of blinding in data analysis could introduce bias, as researchers were aware of group membership during analysis. Future studies could address this limitation by ensuring that data is anonymized until after analysis is completed. Additionally, the study did not control for the adaptation effect [66], which could have influenced the results. This limitation was particularly relevant for data collected during the COVID-19 pandemic, when resources for data collection had to be distributed in advance. Future research could mitigate this by conducting in-person sessions to better control for the adaptation effect.

The Rainbow Passage may not be the optimal paradigm for isolating the effects of grammatical complexity and sentence length on pausing. Although we classified sentences according to established criteria, the passage was not designed to systematically control syntactic structure or to orthogonally manipulate sentence length and complexity. As a result, most simple sentences were short, with only one notable exception. We chose to examine both variables because they capture related but distinct aspects of speech production: sentence length reflects planning demands associated with producing longer sequences of words or syllables, whereas sentence complexity reflects hierarchical syntactic processing and higher-level linguistic planning. The overlap between these variables means that some observed effects on pause number and duration may not be fully independent. Nevertheless, analyzing both provides complementary insight into how different linguistic and production constraints contribute to pausing in PD. Sensitivity analyses indicate that the overall patterns—more frequent and longer pauses in PD—are robust, but we acknowledge that the overlap limits the ability to fully disentangle the separate contributions of length and complexity. In addition, the sentences may not be fully balanced with respect to other articulatory, prosodic, or lexical factors, and the unequal number of simple versus complex sentences further constrains direct comparisons. Future studies using experimentally controlled stimuli that systematically vary sentence length and complexity independently would allow for a more precise evaluation of their individual and interacting effects on pausing.

PD group included fewer female than male participants. Although this imbalance is consistent with epidemiological reports indicating a higher diagnosis rate in males [67], it may also reflect sex-related differences in symptom presentation or health-seeking behaviors that contribute to underdiagnosis in females [68]. Recruitment was further constrained by the requirement that participants remain off dopaminergic medication for 12 hours, which limited enrollment primarily to individuals with mild to moderate disease severity and precluded additional stratification by sex or disease duration. Future studies with larger and more diverse samples are needed to better examine potential sex-related effects in PD speech and pausing behaviour.

Although cognitive status was partially controlled by restricting the range of MoCA scores in the PD group, comprehensive neuropsychological assessments were not conducted. In the current study, correlations were examined between MoCA scores and pause characteristics, as well as between dysarthria severity and pauses, allowing for initial exploration of cognitive and motor contributions. However, more detailed assessment of specific cognitive domains—such as attention, working memory, executive function, and language processing—would help to better dissociate speech motor impairments from cognitive-linguistic influences on pausing. Future work may also benefit from designing speech tasks that more precisely isolate motor versus cognitive contributions to speech timing. In addition, standardized measures of overall motor severity, such as the Movement Disorders Society – Unified Parkinson's Disease Rating Scale (MDS-UPDRS; [69]) and the Hoehn and Yahr staging system [70], were not collected. While participants' ability to tolerate a 12-hour medication withdrawal suggests predominantly mild to moderate motor symptom severity, the absence of formal

motor ratings limits the ability to directly relate pausing behaviour to global motor impairment. Future studies should incorporate comprehensive motor assessments alongside speech measures.

Moreover, data were collected using different formats across sites, with in-person recordings at McGill University and online data collection at the University of Reading due to COVID-19 restrictions. Although identical experimental protocols were followed to the extent possible, differences in recording environments may have introduced variability. Future research would benefit from fully standardized data collection methods or from explicitly modeling recording modality as a factor in the analysis.

Additionally, research could examine whether silent pauses are consistent across different speech subtypes of Parkinson's disease—such as prosodic, phonatory-prosodic, and articulatory-prosodic patterns—potentially contributing to the development of diagnostic screening tools [71].

## Conclusion

The findings of this study suggest that both sentence complexity and length significantly influence the production of silent pauses in PD speech. Clinically, these findings indicate that silent pause duration could serve as an objective measure of dysarthria severity and treatment outcomes. Further research is needed to explore whether silent pause duration could provide complementary information to existing clinical measures, such as intelligibility and the functional impact of dysarthria, in prioritizing patients or assessing treatment outcomes.

## Supporting information

**S1 File. Appendices.** Appendix 1: Criteria for Complexity of Sentence in Rainbow Passage. Based on Dede and Salis (2020). Appendix 2: Categorisation of Sentences in Rainbow Passage.
(DOCX)

**S1 Table. Demographic information of Parkinson group.**
(DOCX)

**S2 Table. Demographic information of Control group.**
(DOCX)

**S3 Table. Praat Coding Criteria.** based on DeDe & Salis (2019) and Reed (2020) – Only showing codes used in from initial criteria.
(DOCX)

**S4 Table. Normality Test Results.**
(DOCX)

## Acknowledgments

We would like to thank Dr. Arpita Bose and Dr. Christos Salis for their technical expertise and help with the data analysis for this project. Additionally, we would like to thank all the participants who took part in this study.

## Author contributions

**Conceptualization:** Fatemeh Mollaei.

**Data curation:** Huw Evans, Alexandra Pool.

**Formal analysis:** Huw Evans, Alexandra Pool.

**Funding acquisition:** Fatemeh Mollaei.

**Investigation:** Fatemeh Mollaei.

**Methodology:** Fatemeh Mollaei, Alexandra Pool.

**Project administration:** Fatemeh Mollaei.

**Resources:** Fatemeh Mollaei.

**Software:** Huw Evans, Alexandra Pool.

**Supervision:** Fatemeh Mollaei.

**Validation:** Fatemeh Mollaei.

**Visualization:** Fatemeh Mollaei.

**Writing – original draft:** Huw Evans, Alexandra Pool.

**Writing – review & editing:** Fatemeh Mollaei.

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
