## [Decision Letter · Decision Letter 0]

26 Nov 2025

Dear Dr. Mollaei,

Thank you for submitting your manuscript to PLOS ONE. After careful consideration, we feel that it has merit but does not fully meet PLOS ONE’s publication criteria as it currently stands. Therefore, we invite you to submit a revised version of the manuscript that addresses the points raised during the review process.

I have now received two reviews for this manuscript, both of which are provide clear, detailed, and actionable feedback. Due to the high quality of the reviews, I will not reiterate their content here, but I encourage the authors to thoroughly address all of the points raised in a revision and a detailed response to reviewers. Provided the authors are willing to do so, I will attempt to send these documents to the same reviewers and ask them to evaluate the extent to which their concerns have been addressed to determine whether the revised manuscript is suitable for publication.

We look forward to receiving your revised manuscript.

Kind regards,

Laura Morett

Academic Editor

PLOS ONE

Journal Requirements:

[This work was supported by a start-up package from the School of Psychology and Clinical Language Sciences at University of Reading awarded to Dr. Fatemeh Mollaei.].

4. In the online submission form, you indicated that [The data sets generated and/or analyzed during the current study are available from the corresponding author upon reasonable request.].

Reviewers' comments:

Reviewer's Responses to Questions

**Comments to the Author**

1. Is the manuscript technically sound, and do the data support the conclusions?

Reviewer #1: Partly

Reviewer #2: Partly

2. Has the statistical analysis been performed appropriately and rigorously?

Reviewer #1: Yes

Reviewer #2: Yes

3. Have the authors made all data underlying the findings in their manuscript fully available?

Reviewer #1: Yes

Reviewer #2: Yes

4. Is the manuscript presented in an intelligible fashion and written in standard English?

Reviewer #1: Yes

Reviewer #2: Yes

Reviewer #1: See attached. Re: item 1 above, the paper is modestly technically sound. The authors need to address questions about the methodology and identify areas of limited rigor such as small sample of female participants.

Reviewer #2: This is an interesting paper that has the potential to make a valuable contribution to the field. The introduction is very well written, but the results and discussion sections would benefit from some further detail and exploration of issues.

My detailed comments are as follows:

Line 157 – I'm not sure I agree with the classification of sentences into long and short. Was this also based on DeDe and Salis? I don’t think a sentence of 17 can be considered short, and one of 18 long just because the happen to lie above and below the mean. The authors should either present what the actual sentence length distribution was, showing that there was a sufficiently large gap between the two categories for the stats to make sense, or re-analyse the data with a different categorisation that results in a sufficient gap if that’s possible.

Line 188 – not sure how a pause below 200 ms would reflect breath?

Line 216 – would it not have been safer to use non-parametric tests under these circumstances?

Line 225 - 227 – this sentence is misleading, I thought you were examining the relationship between cognition and dysarthria until I crosschecked with the research questions.

Line 241 – would be useful to see the actual MLU mean and range here

Line 252 – you already said that in line 247?

Line 279 - slightly confused as you already reported group differences above, e.g. line 268- how is this different?

Line 301 – again these results seem to have already been reported above, see e.g. line 273. Are these tests done for each group? It’s confusing as in the RQ1 section you already refer to individual groups rather than the whole data. I suggest you just have one results section, presenting both ANOVA and any post-hoc tests done on individual group data and tease the results apart according to RQ in the discussion rather than repeating yourself.

Or make it clearer what’s new in the sections that follow the initial presentation and remove any doubled up results from the relevant part.

Overall it’s quite difficult to follow all the descriptive data and stats results, could you present them in a summary table?

Line 331 – highlight again in the text that this refers only to the PD group as it’s easily missed if you don’t read the long heading

Line 403 – so did you have any evidence of this happening in individuals? Pauses are generally excessively long if there is an initiation problem, rather than a few hundred milliseconds.

Line 405 – so were all simple sentences short then?

Line 407 – fewer pauses in which context/sentence type? Or in general?

Line 408 – you might want to consider here the effects of uncontrolled speeding up and decreasing range of movement, i.e. PwPD have problems pacing themselves and stopping which could potentially cause this effect

Line 410 – stats between PD and controls or different sentence types?

Line 415 could you separate your discussion of between sentence and between group effects?

Line 419 – so some research suggests more frequent pauses then, contrasting with Line 407? This needs further explanation if this is based on the speaking task etc.

Line 422 – it’s not really a contradiction, your controls just didn’t have cognitive decline, so you wouldn’t expect that effect.

Line 426 – the paragraph would benefit from a definition of constituent length vs information load, you seem to be mixing the two up further down in line 430, or at least assume that they are related to each other.

Line 437 – why? This needs more detail. This whole paragraph, combined with the next one needs, further exploration – your work has the potential to contribute to the discussion of whether pausing behaviour reflects motor speech difficulties or sentence processing, i.e. cognitive issues, but this is not really explored here. I suggest you look at some of the literature focusing on language difficulties in PwPD and what they found re pausing behaviour to go into more detail here.

Line 445 – did Hammen and Yorkston only refer to speech initiation? There are plenty of other reasons this could be happening

Line 446 - don’t follow this statement, where does speech rate suddenly come from? And why wasn’t that measured if it’s important, doesn’t take that long to extract.

Line 451 – how to you expect to blind the experimenter to group membership when they can hear that the person has a speech disorder?

Line 453 – adaptation would only be an issue if the participant sat and practised reading the passage several times before you recording – how likely was that? Normally you would ask a person to read through a passage silently before reading aloud to avoid any reading issues affecting speech performance anyway, so I don’t think this is a big issue.

Line 458 – don’t understand what the issue is here, what are speech initiation measurements? And did you use a script without checking for accuracy afterwards?

Line 461 – if this is an issue I suggest you go over your stats results again that were not significant to see whether there was any potential for differences without the correction and discuss this in the text.

Line 464 – You need to specify which one of your observations this is based on

Line 467 – this seems a bit far fetched unless you have some references supporting this or provide some more detail for why you think this might be the case?

Line 470 – how do you separate cognitive demands and sentence complexity?

Line 474 – are you actually suggesting that silent pause measures would be more valid to prioritise patients than e.g. their intelligibility, impact of their dysarthria etc.?

Line 476 – and what would that look like in terms of actual treatment provided? I would avoid such generic statements unless they have actually been explored in the text a bit more.

.

Reviewer #1: No

Reviewer #2: No

---

## [Author Response · Author response to Decision Letter 1]

5 Jan 2026

Manuscript ID: PONE-D-25-42565

Responses to the reviewers’ comments

We thank the two reviewers and the Editors of PLOS ONE for the time and effort devoted to evaluating our manuscript. We are grateful for the constructive and insightful comments, which have been carefully considered in revising our work. We appreciate the opportunity to resubmit our manuscript. In response to the reviewers’ suggestions, we have incorporated the requested additional information, provided detailed point-by-point responses to all comments, and thoroughly revised the manuscript. Major changes have been clearly highlighted in the revised version in red. We remain fully committed to addressing any further concerns raised by the reviewers or editors and to correcting any inadvertent errors. Below, we restate each comment followed by our detailed response.

Reviewer #1: See attached. Re: item 1 above, the paper is modestly technically sound. The authors need to address questions about the methodology and identify areas of limited rigor such as small sample of female participants.

Response: Thank you for this observation. We acknowledge that the number of female participants was smaller than that of male participants. Recruiting individuals with Parkinson’s disease who are able to remain off medication for 12 hours is particularly challenging; therefore, we did not further restrict enrollment based on sex. Additionally, Parkinson’s disease is more commonly diagnosed in males than females, with an approximate ratio of 2:1 (Van Den Eeden et al., 2003). However, it remains unclear whether this disparity reflects true differences in disease prevalence or sex-related differences in symptom presentation and health-seeking behavior, which may lead to underdiagnosis in females. Addressing these possibilities would require further investigation beyond the scope of the present study (Miller & Cronin‐Golomb., 2010).

Attachment:

This manuscript focused on pausing behaviors in people with Parkinson’s disease. In general, this is an interesting topic and relevant to improving the development and understanding of speech-based biomarkers in people with PD. The concept behind the study is strong, and it could add to the several papers that have been published about the mechanisms and clinical implications of pausing during speech in PD. However, the authors do not adequately address existing literature and the synthesis and interpretation of their findings is weakened as a result. There are also some inconsistencies and areas of decreased clarity in the methodology.

Introduction:

The authors touch on several unmet needs in understanding and treating speech dysfunction in people with PD, but the topics are somewhat tangential. The connection between speech-based markers and impact on clinical practice could be made more direct and clearer. The authors could make a stronger case for why pause characterization may improve evaluation and treatment. There are many studies about pause timing in PD, so the statement that less attention has been paid is not well supported (see several Huber et al. papers, Smith et al. 2018, Andrade et al. 2023 among others).

Lines 83-87 would be strengthened by reviewing this literature and explaining the current state of knowledge about speech rate, articulation rate, and pause behaviors with more up-to-date literature. It may be helpful to define these terms and how they relate to pause measures and their implementation in research paradigms.

Response: We thank the reviewer for bringing this point. We have now revised the Introduction to include a more direct link between the pause characteristics and the evaluation treatment of speech in PD. Please see below and also in the manuscript lines 80-99:

“In PD, speech timing is commonly disrupted, with pause behaviour emerging as a more consistent feature than changes in speech or articulation rate. Speech rate refers to the overall speed of speech production, typically measured as syllables or words per second including pauses, whereas articulation rate reflects the speed of articulatory movements excluding pauses, and pause timing encompasses the frequency, duration, and placement of silent intervals during speech. Speech rate is often reduced in PD, although findings are mixed and influenced by task type and disease stage, while articulation rate appears relatively preserved, suggesting that articulatory execution is less affected than higher-level planning and timing processes (Blanchet & Snyder, 2009; Illner et al., 2022; Moreau & Pinto, 2019). In contrast, pause duration and distribution constitute robust markers of PD speech, reflecting dysfunction within basal ganglia–cortical circuits involved in speech sequencing and timing, as supported by evidence that deep brain stimulation modulates pause characteristics (Ahn et al., 2014; Guenther, 2016). Abnormal pausing may also reflect linguistic and cognitive difficulties, including lexical retrieval, syntactic processing, and executive control, and has been linked to cognitive function in PD, highlighting its potential value as a speech-based marker of cognitive decline (Bohsali & Crosson, 2016; Lee et al., 2019; Andrade et al., 2023). Clinically, these findings suggest that assessment and management of speech in PD should move beyond global speech rate measures to include detailed analysis of pause timing, particularly in connected speech, and that interventions focused solely on slowing speech may be less effective than approaches targeting planning, prosodic phrasing, and respiratory–speech coordination (Knowles et al., 2021; Lowit et al., 2018; for a review, see Fumel et al., 2024).”

Methods:

Later parts of the paper discuss syllable content (lines 426-432) but this is not mentioned in the methods as a criteria used to categorize the sentence types. Please clarify.

Response: This paragraph describes the variables under investigation, namely sentence complexity (simple vs. complex) and sentence length (short vs. long; see Appendix 2 for the classification criteria). The number of syllables is used as an index of sentence length, with longer sentences containing a higher syllable count. To improve clarity, we have rephrased this paragraph accordingly. As noted at the end of the paragraph, syllable count was not explicitly controlled for in the present study and may therefore represent a potential confound that warrants further investigation please refer to lines 442-451.

The authors should state their rationale for pause duration thresholds.

Response: As we pointed in the manuscript pauses were set at a boundary of 200 ms, based on recommendations by DeDe and Salis (2020) to capture shorter pauses while excluding those that likely corresponded to breath or articulation. We have expanded here to include the rationale for pause duration threshold clear. Please see below and refer to lines 208-224:

“A TextGrid was created in Praat to mark silent pauses, defined as intervals exceeding 150 ms based on (Alvar et al., 2019). For this study, pauses were set at a boundary of 200 ms, based on recommendations by DeDe and Salis (2020), to capture shorter pauses while excluding those that likely corresponded to breath or articulation. A 200-ms pause duration threshold is commonly justified in studies of PD speech because it provides a principled boundary between brief articulatory gaps and functionally meaningful pauses related to speech planning, timing, and cognitive–linguistic processing (Lee et al., 2019). Silent intervals shorter than approximately 200 ms are generally considered part of normal articulatory coordination and segmental timing, whereas pauses exceeding this duration are more likely to be perceived by listeners as disruptions to speech flow and to reflect higher-level processes such as lexical retrieval, syntactic planning, or respiratory control (Goldman-Eisler, 1968; Campione & Véronis, 2002). In PD, basal ganglia dysfunction is known to affect speech timing and sequencing, making pauses above this threshold particularly relevant as markers of impaired motor planning and cognitive–linguistic integration (Guenther, 2016; Ahn et al., 2014).Using a 200-ms cutoff therefore helps isolate pauses that are sensitive to PD-related speech impairments while excluding micro-silences that are unlikely to be clinically or theoretically meaningful, and this threshold has been widely adopted in analyses of connected speech in PD and related populations (Campione & Véronis, 2002; Lee et al., 2019).”

Importantly, simple/complex and short/long sentence categorizations are not mutually exclusive. This fact makes the findings more difficult to interpret.

Response: Thanks for this point. We agree that considering the complexity and length of sentences makes the findings more difficult to interpret, and that’s why we have compared the duration and the number of pauses across sentence complexity (simple vs. complex) and sentence length (short vs. long) as two factors and interpreted our results for each one separately as well (Research Question 2: Does sentence complexity and length affect pause behaviour?)

Please state how many raters there were.

Response: This now has been added to the manuscript, see line 241:

“Inter-rater reliability was assessed using qualitative ratings for the number of pauses and duration of pauses by two independent raters.”

Discussion:

There are missing topic areas that would add to the synthesis of findings and place their work in the greater context of the literature. One is respiratory pauses and another is grammatical pauses. The paper would be stronger with a more well-conceived hypothesis to explain their findings in light of these different pause behaviors and their mechanisms. Finally, speech task type is not addressed at all. Mention of prior studies should specify if the speech task was reading or spontaneous speech, because studies done using spontaneous speech would understandably demonstrate inconsistencies compared to this paper using reading only. Notably, lexical retrieval and word selection (line 436) do not apply to read speech.

Response: Thanks for these thoughtful points that will undoubtedly improve the Discussion and position it in the greater context. We have now revised the Discussion to include the respiratory pauses and grammatical pauses. In addition, we have included on the speech task type comparing reading to other types of tasks that previous studies have used. We have also removed this sentence on line 436 from the revised version of the manuscript: “These results align with expectations, suggesting that cognitive ability influences the efficiency of lexical retrieval and word selection in PD speech”. Please refer to lines 455-477 in the revised version of the manuscript and see below:

“Previous research has shown that individuals with PD exhibit longer and more frequent silent pauses, reduced use of filled pauses, and poorer alignment of respiratory pauses with grammatical boundaries, reflecting disruptions in linguistic planning, automatic cueing, and respiratory–speech coordination (Rosen et al., 2010; Huber & Darling-White, 2017; Lowit et al., 2018). Much of this evidence comes from spontaneous or semi-spontaneous speech tasks, such as narratives or picture descriptions, which place high demands on lexical retrieval, discourse planning, and internal timing, thereby amplifying pause abnormalities in PD (Rosen et al., 2010; Ash et al., 2012). Silent pauses in PD have been associated with increased cognitive–linguistic load and lexical access difficulty, whereas filled pauses—typically serving as planning buffers—are reduced, likely due to basal ganglia dysfunction affecting internally generated cues (Rosen et al., 2010; Ash et al., 2012). This study extends this literature by examining pausing during a controlled reading task (Rainbow Passage), in which lexical content and syntactic structure are externally provided, reducing discourse-level planning demands while preserving sentence-level formulation and respiratory coordination requirements. Despite these reduced demands, sentence complexity (simple vs. complex) and sentence length (short vs. long) systematically modulated pause behavior in PD, with longer and more complex sentences eliciting longer silent pauses, particularly within sentences. Although respiratory pauses were not directly measured, the altered distribution of silent pauses within grammatical structures is consistent with prior findings of impaired coordination between respiration and linguistic planning in PD (Lowit et al., 2018). Together, these findings indicate that pause abnormalities in PD are evident across speech task types and reflect an interaction between motor–respiratory constraints and higher-level linguistic planning, with sentence complexity and length exacerbating these effects even during passage reading.”

There is no discussion of any detected patterns concerning filled and unfilled pauses, but this would make the findings more interesting and help explain the behavioral driving factors of these pauses in the PD group.

Response: We agree with this point and have added the discussion on filled and silent/unfilled pauses with the pervious point that you brought up. Please see the response above and refer to the revised version of the manuscript lines 455-477.

In general, in a few places the authors note “disease severity” when “dysarthria severity” is really what can be measured under their methodology.

Response: Thank you for this comment. We agree that our measurements reflect dysarthria severity rather than overall disease severity. We have revised the manuscript accordingly and replaced all instances of “disease severity” with “dysarthria severity.”

Additional limitations include the small number of females in the sample, and wide disease duration range, and the use of MoCA as cognitive categorization practice. Importantly, there are no motor symptom measures and there seem to be differences between the data collection procedures at the two sites.

Response: Thanks for these thoughtful comments. We acknowledge that the number of female participants in the present study was smaller than the number of male participants. Recruitment of individuals with Parkinson’s disease (PD) who are able to remain off dopaminergic medication for 12 hours is particularly challenging and tends to limit enrollment to individuals with mild to moderate disease severity. For this reason, we did not further restrict participation based on sex or disease duration. In addition, PD is more frequently diagnosed in males than in females, with an approximate male-to-female ratio of 2:1 (Van Den Eeden et al., 2003). However, it remains unclear whether this disparity reflects true differences in disease prevalence or sex-related differences in symptom presentation and health-seeking behaviors that may contribute to underdiagnosis in females, an issue that warrants further investigation beyond the scope of the present study (Miller et al., 2010).

To reduce potential confounds related to cognition, we controlled for the range of MoCA scores within the PD group, indicating that participants did not exhibit severe cognitive impairment that could influence speech performance beyond PD-related effects. We did not collect standardized measures of overall motor severity (e.g., UPDRS or Hoehn and Yahr staging); however, given that participants were able to tolerate a 12-hour medication withdrawal, we infer that the sample primarily consisted of individuals with mild to moderate motor symptom severity. We acknowledge the absence of formal motor severity measures as a limitation and note that future studies would benefit from incorporating comprehensive assessments of motor symptoms beyond speech.

With respect to data collection procedures, we acknowledge that data were collected in different formats (in person at McGill University and online at the University of Reading due to COVID-19 restrictions). Nevertheless, we took care to ensure that identical protocols were followed across sites. These methodological considerations and limitations have now been explicitly acknowledged in the Limitations and future directions section of the manuscript (lines 518-544) and below:

“PD group in

---

## [Decision Letter · Decision Letter 1]

5 Feb 2026

Dear Dr. Mollaei,

Thank you for submitting your manuscript to PLOS ONE. After careful consideration, we feel that it has merit but does not fully meet PLOS ONE’s publication criteria as it currently stands. Therefore, we invite you to submit a revised version of the manuscript that addresses the points raised during the review process.

I thank the authors for thoroughly addressing the points raised by the reviewers in the first round of review. Although most of the remaining points raised are minor, R2 raises one point concerning the length and complexity of sentences that warrants either reconsideration of the analyses used or strengthening of the argument for the inclusion of this variable, hence the major revision recommendation. I encourage the authors to submit another revision addressing these remaining points. Should they decide to do so, I will attempt to return the manuscript to the same reviewers once again to ensure that all remaining points have been addressed.

We look forward to receiving your revised manuscript.

Kind regards,

Laura Morett

Academic Editor

PLOS One

Journal Requirements:

Reviewers' comments:

Reviewer's Responses to Questions

**Comments to the Author**

Reviewer #1: (No Response)

Reviewer #2: All comments have been addressed

2. Is the manuscript technically sound, and do the data support the conclusions?

Reviewer #1: Yes

Reviewer #2: Partly

3. Has the statistical analysis been performed appropriately and rigorously?

Reviewer #1: Yes

Reviewer #2: Yes

4. Have the authors made all data underlying the findings in their manuscript fully available?

Reviewer #1: Yes

Reviewer #2: Yes

5. Is the manuscript presented in an intelligible fashion and written in standard English?

Reviewer #1: Yes

Reviewer #2: Yes

Reviewer #1: The authors have done very well in their revisions, and have thoroughly and thoughtfully addressed most of my comments. The manuscript is substantially improved. Additional points I am hoping the authors can clarify in the discussion are:

1. In Lines 399-404, I do not understand how references 41 and 55 are being interpreted in the context of the findings just presented. Are the authors saying that both papers support that both cognitive and motor function are associated with fluency and pausing, or is the intended meaning something more specific? In particular, the last line at 403-404 is confusing. I do not know what "factors" need to be investigated further.

2. The section on speech acceleration (Lines 420-430) would benefit from some additional clarification. My impression was that in this study, PD participants globally had an increased number and duration of pauses compared to older controls, so I am not sure what data supports festination or acceleration of speech timing in this present work.

3. Looking more closely at the single/complex sentence findings, I would consider adding to your discussion the possibility that the rainbow passage was not the optimal experimental paradigm to assess grammatical complexity and pausing, thus further research is needed. I understand the approach used to classify the sentences, but it is difficult to know how well balanced these sentences may be from other articulatory and linguistic perspectives, and also there is not an even number of simple vs. complex.

Reviewer #2: PONE comments

Thank you to the authors for their careful consideration of the reviewer comments, generally I am happy that the issues have been addressed and I only have some minor comments on the revised material as outlined below. However, one of the responses re. sentence length/complexity has opened up a new area of debate with regard to research question 2 which requires more detailed consideration, again see comment below.

L73-75 – two sentences don’t work together – one is on management, the other on speech characteristics

L 85-88 I appreciate there are reference to support this statement, however, the description here is too one-sided , i.e. 1, the authors should also mention the fact that PwPD might have accelerated rate, and 2, there are definitely articulatory problems, such as undershoot, maybe they just don’t feature as much in milder patients.

L88-91 more evidence needed that pausing behaviour is more robust beyond a DBS study

L91-94 some repetition here and in the next section

L150 sorry if I missed this but what was the dysarthria severity rating based on?

L265 the results section is really difficult to read with all the stats results in the text. They seem redundant given that a summary is provided in Table 1, so I would suggest removing them from the text. I also wonder whether it would be more logical to report the post-hoc results in the same paragraph as the ANOVA so that they are grouped by variable rather than statistical analysis method, but I leave that up to the authors to decide.

Previous L405 (see response to reviewer 2): I do not agree that the long sentence you refer to here is a simple sentence. Whilst there is no clear subordinate clause with a new verb here, the prepositional phrase starting with “with…” is not the simple phrase you normally encounter, such as in “I saw a man with a dog”. In fact, you could reinsert a verb “with it’s path being high above” which clearly denotes it as a subordinate clause. Whether you agree with this analysis or not, the sentence definitely stands out from the rest of your simple sentences and I would therefore check whether the results somehow bias the rest of the set and potentially exclude it from analysis as an anomaly.

Next, having established that with the exception of that one sentence, your simple sentences all fall into the short category and complex one in the long category, what exactly is the point of analysing both of these variables? If you wanted to differentiate between more/longer pauses being caused by higher level linguistic processing constraints versus speech production limitations, then this should have been controlled better by also including short complex sentences, or long, simple ones (e.g. item lists) which is obviously difficult to do in a standard reading passage. I would therefore like to see a much clearer argument for why both variables were considered and an exploration of the extent to which the results were influenced by the overlap in the discussion if it is decided that they should both remain.

L516 – doesn’t make sense to me, two different raters putting the same speech samples through the same script should not show any differences, Praat does not change its analysis just because another rater presses the run button. What exactly was being examined here?

L527 above you have included a correlational analysis of dysarthria severity with MoCA scores, so clearly you did more than set a score range as an inclusion criterion. Also, which particular cognitive features would you suggest to investigate to differentiate speech from cognitive effects? I think you might also need to rethink your speech task if this an aim.

L553 be specific, only pause duration showed any link to dysarthria severity

.

Reviewer #1: No

Reviewer #2: No

---

## [Author Response · Author response to Decision Letter 2]

18 Mar 2026

Manuscript ID: PONE-D-25-42565R1

Responses to the reviewers’ comments

We thank the reviewers and Editors of PLOS ONE for their continued time and constructive feedback. In response to the second round of review, we have carefully considered each point and made further revisions to improve clarity and completeness, which are highlighted in red in the revised manuscript. Detailed point-by-point responses are provided below, and we remain committed to addressing any additional concerns to strengthen the manuscript.

Reviewer #1: The authors have done very well in their revisions, and have thoroughly and thoughtfully addressed most of my comments. The manuscript is substantially improved. Additional points I am hoping the authors can clarify in the discussion are:

1. In Lines 399-404, I do not understand how references 41 and 55 are being interpreted in the context of the findings just presented. Are the authors saying that both papers support that both cognitive and motor function are associated with fluency and pausing, or is the intended meaning something more specific? In particular, the last line at 403-404 is confusing. I do not know what "factors" need to be investigated further.

Response: Thanks for this thoughtful comment, we understand that these sentences may be confusing. We have now re-phrased them to read as below. Please see lines 391-398 in the revised manuscript:

“It has been found that pause duration correlates with executive functioning, grammatical ability, and motor initiation, indicating that linguistic planning and executive control — not just motor capacity — influence how long speakers pause [23]. Krivokapić [43] also suggests that sentence length may influence both pre- and post-boundary pause durations, which increases processing effort and planning time. This could explain the observed increase in between-sentence pauses. However, future studies need to investigate linguistic, motor, and cognitive factors to better understand their relative contributions to pause durations.”

2. The section on speech acceleration (Lines 420-430) would benefit from some additional clarification. My impression was that in this study, PD participants globally had an increased number and duration of pauses compared to older controls, so I am not sure what data supports festination or acceleration of speech timing in this present work.

Response: Thank you for this comment. We have now clarified this point to mention that those previous findings on festination and acceleration of speech timing are at odds with our findings and provided a possible explanation for it with more clarity. See blow and lines 419-424 in the revised manuscript:

“These findings are inconsistent with ours. The reason for this discrepancy remains unclear. However, one possible explanation is that uncontrolled acceleration of speech (speech festination) and difficulty terminating ongoing motor actions—both of which may occasionally occur in individuals with PD speech—could lead to compressed speech characterized by fewer, or even absent, planned pauses, particularly at linguistically appropriate boundaries [22,59]. However, we did not observe this pattern in our own data.”

3. Looking more closely at the single/complex sentence findings, I would consider adding to your discussion the possibility that the rainbow passage was not the optimal experimental paradigm to assess grammatical complexity and pausing, thus further research is needed. I understand the approach used to classify the sentences, but it is difficult to know how well balanced these sentences may be from other articulatory and linguistic perspectives, and also there is not an even number of simple vs. complex.

Response: Thank you for bringing up this point. We have added a paragraph in the Limitation and future directions section that addresses this comment. Please see blew and lines 51--529 in the revised manuscript:

“The Rainbow Passage may not be the optimal paradigm for isolating the effects of grammatical complexity and sentence length on pausing. Although we classified sentences according to established criteria, the passage was not designed to systematically control syntactic structure or to orthogonally manipulate sentence length and complexity. As a result, most simple sentences were short, with only one notable exception. We chose to examine both variables because they capture related but distinct aspects of speech production: sentence length reflects planning demands associated with producing longer sequences of words or syllables, whereas sentence complexity reflects hierarchical syntactic processing and higher-level linguistic planning. The overlap between these variables means that some observed effects on pause number and duration may not be fully independent. Nevertheless, analyzing both provides complementary insight into how different linguistic and production constraints contribute to pausing in PD. Sensitivity analyses indicate that the overall patterns—more frequent and longer pauses in PD—are robust, but we acknowledge that the overlap limits the ability to fully disentangle the separate contributions of length and complexity. In addition, the sentences may not be fully balanced with respect to other articulatory, prosodic, or lexical factors, and the unequal number of simple versus complex sentences further constrains direct comparisons. Future studies using experimentally controlled stimuli that systematically vary sentence length and complexity independently would allow for a more precise evaluation of their individual and interacting effects on pausing.”

Reviewer #2: PONE comments

Thank you to the authors for their careful consideration of the reviewer comments, generally I am happy that the issues have been addressed and I only have some minor comments on the revised material as outlined below. However, one of the responses re. sentence length/complexity has opened up a new area of debate with regard to research question 2 which requires more detailed consideration, again see comment below.

L73-75 – two sentences don’t work together – one is on management, the other on speech characteristics

Response: Thanks for this comment. We have revised the second sentence. Please see lines 74-77 in the revised version of the manuscript, and below:

“However, these approaches primarily focus on global speech outcomes, and a more detailed understanding of how PD affects specific components of speech—such as pause behaviour—is still limited. This gap may constrain the development of more targeted and mechanism-based intervention strategies.”

L 85-88 I appreciate there are reference to support this statement, however, the description here is too one-sided , i.e. 1, the authors should also mention the fact that PwPD might have accelerated rate, and 2, there are definitely articulatory problems, such as undershoot, maybe they just don’t feature as much in milder patients.

Response: We appreciate this comment, and have revised the manuscript to acknowledge the accelerated rate and articulatorily problems in PD. Pease see below and refer to the lines 87-92 in the revised manuscript:

“Speech rate in PD is frequently reported as reduced, although findings are mixed and influenced by task type and disease stage; in some cases, individuals with PD may also exhibit an accelerated or festinating speech rate [19-21]. While articulation rate is often described as relatively preserved, particularly in milder stages, articulatory impairments such as reduced movement amplitude and articulatory undershoot may also be present [22].”

L88-91 more evidence needed that pausing behaviour is more robust beyond a DBS study

Response: Thank you for this comment. We have added more references to support this statement. Please see below and lines 92-98.

“Increased frequency and duration of pauses have been observed across a range of speech tasks, including spontaneous speech and reading, and have been associated with reduced speech fluency and executive-linguistic demands [e.g., 19–21, and 23–26]. These findings are further supported by evidence that modulation of basal ganglia–cortical circuits through deep brain stimulation can influence pause characteristics [27, 28:280–283], suggesting that pausing behaviour may reflect disruption of neural systems involved in speech sequencing and timing.”

L91-94 some repetition here and in the next section

Response: Thank you for this observation. We have revised this section and removed this sentence. Please refer to lines 99-106 in the revised version of the manuscript, and see below:

“Together, these findings suggest that both higher-level timing processes and segmental articulatory control can be affected in PD, with their relative contributions varying by individual and disease severity. Altered pause duration and distribution are consistently reported and may represent a clinically relevant feature of PD speech. These observations highlight the importance of moving beyond global speech rate measures toward more detailed analyses of pause timing in connected speech, and suggest that interventions targeting planning, prosodic phrasing, and respiratory–speech coordination may be beneficial [29, 30; see also 31].”

L150 sorry if I missed this but what was the dysarthria severity rating based on?

Response: Thank you for pointing this out. The dysarthria severity rating was already included in the original manuscript and was based on the 7-point scale by Duffy (2005). Please refer to the Procedure section of the Materials and Methods (lines 243-246) for the full details and below:

“Passage recordings were additionally analyzed for clinical dysarthria severity, rated by a trained Speech and Language Therapist (SLT) using a 7-point scale (1 = normal speech, 7 = severe speech) based on perceptual characteristics related to articulation, resonance, prosody, phonation, and respiration [54].”

L265 the results section is really difficult to read with all the stats results in the text. They seem redundant given that a summary is provided in Table 1, so I would suggest removing them from the text. I also wonder whether it would be more logical to report the post-hoc results in the same paragraph as the ANOVA so that they are grouped by variable rather than statistical analysis method, but I leave that up to the authors to decide.

Response: We thank the reviewer for this helpful suggestion. We have revised the Results section to improve clarity and readability. Specifically, we reduced the amount of detailed statistical reporting in the main text, retaining only the direction and significance of effects while referring readers to Table 2 for full statistical details. In addition, we reorganized the section to group ANOVA and post-hoc findings by outcome variable (pause number and pause duration) rather than by statistical procedure. We believe this restructuring substantially improves the logical flow and accessibility of the Results section while preserving transparency of reporting. Please see lines 276-321 in the revised version of the manuscript.

Previous L405 (see response to reviewer 2): I do not agree that the long sentence you refer to here is a simple sentence. Whilst there is no clear subordinate clause with a new verb here, the prepositional phrase starting with “with…” is not the simple phrase you normally encounter, such as in “I saw a man with a dog”. In fact, you could reinsert a verb “with it’s path being high above” which clearly denotes it as a subordinate clause. Whether you agree with this analysis or not, the sentence definitely stands out from the rest of your simple sentences and I would therefore check whether the results somehow bias the rest of the set and potentially exclude it from analysis as an anomaly.

Response: We thank the reviewer for this careful reading. We agree that the sentence “These take the shape of a long, round arch, with its path high above and its two ends apparently beyond the horizon” is atypical among the simple sentences and contains a complex prepositional phrase that could be interpreted as a reduced subordinate clause. However, according to our predefined criteria for simple versus complex sentences—where a simple sentence contains a single independent clause and may include compound nouns or verbs, and a complex sentence requires at least one dependent clause introduced by a subordinating conjunction or relative pronoun—this sentence meets the criteria for a simple sentence, as it has only one independent clause and no full subordinate clause with an explicit verb.

Importantly, to ensure that this atypical sentence does not bias the results, we conducted a sensitivity analysis excluding it. Re-running the ANOVAs for pause number and duration with this sentence removed did not materially change the pattern or significance of the results (all main effects and interactions reported in Table 2 remained significant). Therefore, while we acknowledge that the sentence is somewhat atypical, it does not appear to influence the overall findings and was retained in the analyses for completeness and consistency with the standardized Rainbow Passage.

Next, having established that with the exception of that one sentence, your simple sentences all fall into the short category and complex one in the long category, what exactly is the point of analysing both of these variables? If you wanted to differentiate between more/longer pauses being caused by higher level linguistic processing constraints versus speech production limitations, then this should have been controlled better by also including short complex sentences, or long, simple ones (e.g. item lists) which is obviously difficult to do in a standard reading passage. I would therefore like to see a much clearer argument for why both variables were considered and an exploration of the extent to which the results were influenced by the overlap in the discussion if it is decided that they should both remain.

Response: We thank the reviewer for this important point. We agree that, in the Rainbow Passage, there is substantial overlap between sentence length and sentence complexity, with most simple sentences being short and most complex sentences being long. Our rationale for analyzing both variables was to capture complementary aspects of speech production: sentence length reflects planning load related to the number of words or syllables, whereas sentence complexity reflects syntactic or hierarchical linguistic processing demands. While the standard Rainbow Passage does not allow for fully crossing length and complexity (e.g., short complex or long simple sentences), examining both variables provides descriptive insight into how these different linguistic and production factors may influence pausing.

To address the reviewer’s concern about overlap, we explored the results with respect to the relationship between length and complexity. The patterns of increased pause number and duration were broadly consistent across analyses, suggesting that both factors contribute to pausing, but we acknowledge that we cannot fully disentangle the effects of length versus syntactic complexity in this dataset. We have added a discussion of this limitation and its implications for interpreting our findings, highlighting that future studies could benefit from experimental designs that orthogonally manipulate sentence length and complexity to better isolate their contributions to pausing. Please see lines 510-529 in the Limitations and future directions section, and see below:

“The Rainbow Passage may not be the optimal paradigm for isolating the effects of grammatical complexity and sentence length on pausing. Although we classified sentences according to established criteria, the passage was not designed to systematically control syntactic structure or to orthogonally manipulate sentence length and complexity. As a result, most simple sentences were short, with only one notable exception. We chose to examine both variables because they capture related but distinct aspects of speech production: sentence length reflects planning demands associated with producing longer sequences of words or syllables, whereas sentence complexity reflects hierarchical syntactic processing and higher-level

---

## [Decision Letter · Decision Letter 2]

8 Apr 2026

Pause characteristics of sentence production in Parkinson's disease: Insights from sentence complexity and length

PONE-D-25-42565R2

Dear Dr. Mollaei,

We’re pleased to inform you that your manuscript has been judged scientifically suitable for publication and will be formally accepted for publication once it meets all outstanding technical requirements.

Kind regards,

Laura Morett

Academic Editor

PLOS One

Additional Editor Comments (optional):

I thank the authors for their attention to the remaining points raised by the reviewers. The reviewers are now satisfied, and I am pleased to recommend the manuscript for publication.

Reviewers' comments:

Reviewer's Responses to Questions

**Comments to the Author**

Reviewer #1: All comments have been addressed

Reviewer #2: All comments have been addressed

2. Is the manuscript technically sound, and do the data support the conclusions?

Reviewer #1: Yes

Reviewer #2: Yes

3. Has the statistical analysis been performed appropriately and rigorously?

Reviewer #1: Yes

Reviewer #2: Yes

4. Have the authors made all data underlying the findings in their manuscript fully available?

Reviewer #1: Yes

Reviewer #2: Yes

5. Is the manuscript presented in an intelligible fashion and written in standard English?

Reviewer #1: Yes

Reviewer #2: Yes

Reviewer #1: (No Response)

Reviewer #2: (No Response)

.

Reviewer #1: No

Reviewer #2: No

---

## [Editor Report · Acceptance letter]

PONE-D-25-42565R2

PLOS One

Dear Dr. Mollaei,

I'm pleased to inform you that your manuscript has been deemed suitable for publication in PLOS One. Congratulations! Your manuscript is now being handed over to our production team.

Kind regards,

on behalf of

Dr. Laura Morett

Academic Editor

PLOS One